# Non-Compositionality in Sentiment: New Data and Analyses

**Verna Dankers** and **Christopher G. Lucas**
Institute for Language, Cognition and Computation
University of Edinburgh
vernadankers@gmail.com, c.lucas@ed.ac.uk

## Abstract

When natural language phrases are combined, their meaning is often more than the sum of their parts. In the context of NLP tasks such as sentiment analysis, where the meaning of a phrase is its sentiment, that still applies. Many NLP studies on sentiment analysis, however, focus on the fact that sentiment computations are largely compositional. We, instead, set out to obtain non-compositionality ratings for phrases with respect to their sentiment. Our contributions are as follows: a) a methodology for obtaining those non-compositionality ratings, b) a resource of ratings for 259 phrases – NON-COMPSST – along with an analysis of that resource, and c) an evaluation of computational models for sentiment analysis using this new resource.

## 1 Introduction

In NLP, the topics of the compositionality of language and neural models' capabilities to compute meaning compositionally have gained substantial interest in recent years. Yet, the meaning of linguistic utterances often does not adhere to strict patterns and can be surprising when looking at the individual words involved. This affects how those utterances behave in downstream tasks, such as sentiment analysis. Given a phrase or sentence, that task involves predicting the polarity as positive, negative or neutral. Sentiment largely adheres to compositional principles (Moilanen and Pulman, 2007, p.1): "If the meaning of a sentence is a *function* of the meanings of its parts then the *global polarity* of a sentence is a *function* of the *polarities* of its parts." Modelling sentiment as a compositional process is, therefore, often mentioned as a design principle for computational sentiment models (e.g. by Socher et al., 2013; Sutherland et al., 2020; Yin et al., 2020).

Nonetheless, one can think of examples where the sentiment of a phrase is unexpected given the

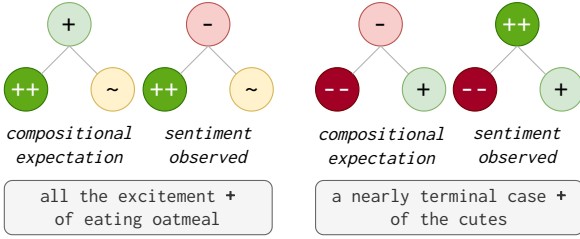

Figure 1: Illustration of how the observed sentiment deviates from the expected sentiment when viewing polarity as a function of the polarity of the subphrases. These examples were obtained from our newly annotated stimuli.

sentiment of the individual parts (e.g. see Zhu et al., 2015; Hwang and Hidey, 2019; Barnes et al., 2019; Tahayna et al., 2022, for work on non-compositional sentiment). These include the case of sarcasm ("life is good, you should get one"), opposing sentiments ("terribly fascinating"), idiomatic expressions ("break a leg") and neutral terms that, when composed, suddenly convey sentiment ("yeah right"). Adequately capturing sentiment computationally requires both learning compositional rules and understanding when such exceptions exist, where most contemporary sentiment models are expected to learn that via mere end-to-end training on examples.

How can we identify whether the sentiment of a phrase is non-compositional? We design a protocol to obtain such non-compositionality judgments based on human-annotated sentiment. Our methodology (elaborated on in §3) utilises phrases from the *Stanford Sentiment Treebank* (SST) (Socher et al., 2013) and contrasts the sentiment of a phrase with control stimuli, in which one of the two subphrases has been replaced. Phrases whose annotated sentiment deviates from what is expected based on the controls are considered less compositional, as is illustrated in Figure 1. We analyse the resulting non-compositionality ranking of phrases

(§4) and show how the constructed resource can be used to evaluate sentiment models (§5). Our new resource (NONCOMPSST) can further improve the understanding of what underlies non-compositionality in sentiment analysis, and can complement existing evaluation protocols for sentiment analysis models.

## 2 Related work

Over the course of years, sentiment analysis systems went from using rule-based models and sentiment lexicons (e.g. Moilanen and Pulman, 2007; Taboada et al., 2011) to using recursive neural networks (Socher et al., 2013; Tai et al., 2015; Zhu et al., 2015), to abandoning the use of structure altogether by finetuning pretrained large language models (e.g. Pérez et al., 2021; Camacho-collados et al., 2022; Hartmann et al., 2023), and recently, to abandoning training, via zero-shot generalisation (Wang et al., 2023). Crucial to the development of these systems has been the introduction of benchmarks, such as SST (Socher et al., 2013) and SemEval's Twitter benchmarks (e.g. Rosenthal et al., 2015; Nakov et al., 2016; Rosenthal et al., 2017).

Although the vast majority of related work focused on simply improving on benchmarks, there have been studies more closely related to ours, asking questions such as: How do phrases with opposing sentiment affect each other (Kiritchenko and Mohammad, 2016a,b)? What is the role of negations (Zhu et al., 2014), modals and adverbs (Kiritchenko and Mohammad, 2016c)? Do idioms have non-compositional sentiment (Hwang and Hidey, 2019)? Which linguistic phenomena are still problematic for SOTA sentiment systems (Barnes et al., 2019)? Can we incorporate compositional and non-compositional processing in one system (Zhu et al., 2015)? And can we computationally rank sentences according to their sentiment compositionality (Dankers and Titov, 2022)? Gaining a better understanding of the contexts in which sentiment functions non-compositionally and is challenging to predict is crucial for the evaluation of sentiment models in an age where sentiment benchmarks may appear saturated (Barnes et al., 2019).[1]

We position our work in this latter group of articles, of which that of Hwang and Hidey is most closely related.

---

[1] As a concrete example, consider the widely-used binary SST sentiment analysis task contained in the GLUE benchmark (Wang et al., 2018): at the time of writing, SOTA performance for this task matches humans' performance.

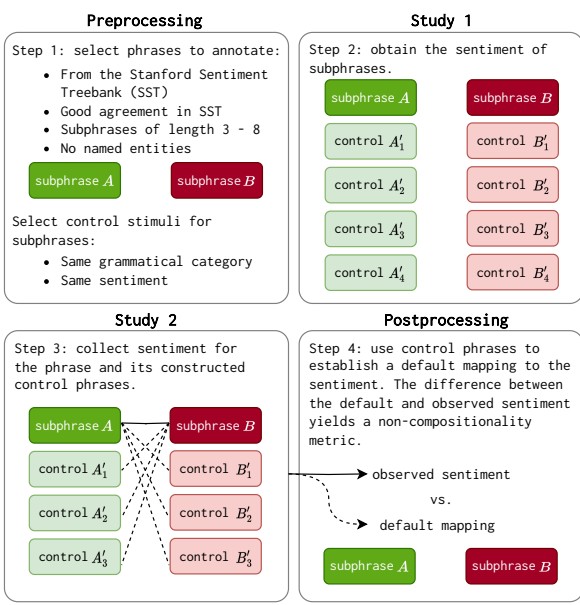

Figure 2: The methodology summarised. Steps 1 and 4 consist of data pre- and postprocessing; steps 2 and 3 involve collecting data from participants via Prolific.

## 3 Collecting non-compositionality ratings

Compositional processing of sentiment involves applying a function to the polarity of subphrases to obtain the polarity of the phrase. Turning this notion into a quantifiable metric requires us to measure the polarities and determine the composition function. Non-compositional phrases are then simply phrases whose sentiment deviates from what is expected. How do we implement this? We first select data (§3.1) and then obtain sentiment labels for phrases through data annotation studies (§3.2). We consider the composition function to be the default mapping from two subphrases with a specific sentiment to their combined sentiment. We obtain the default mapping by replacing subphrases with control stimuli and annotating sentiment for those modified phrases. Using those results, we can compute the non-compositionality ratings (§3.3). Figure 2 summarises the full procedure.

### 3.1 Materials

We first select phrases for which to obtain the non-compositionality ratings, along with control stimuli.

**Data selection** We obtain our data from the SST dataset, containing 11,855 sentences from movie reviews (Pang and Lee, 2005), and sentiment annotations from Socher et al. (2013). The dataset provides sentiment labels for all full sentences and

phrases contained in these sentences (all phrases that represent a node in the constituency parse trees of these sentences). We select candidate phrases to include in our dataset by applying the following constraints to the phrases: they consist of two subphrases that contain 3-8 tokens each, do not contain named entities and had a relatively high agreement in the original dataset. In Appendix A, we elaborate on the implementation of our constraints.

**Selection of control subphrases** We assume that if a subphrase behaves compositionally, replacing it with a control should not affect the overall sentiment of the phrase. How do we select control stimuli? By taking subphrases with the same sentiment (based on SST's sentiment labels) and phrase type (e.g. NP, PP, SBAR). For each phrase – consisting of subphrases $A$ and $B$ – we automatically select 32 candidate control subphrases and manually narrow them down to eight ($A'_n$, $B'_n$, where $n \in \{1, 2, 3, 4\}$). During the manual annotation, we removed examples for which fewer than eight suitable control stimuli remained. Our final collection contains 500 phrases to be used in the human annotation study.

### 3.2 Data annotation studies

We collect sentiment labels in two rounds using a 7-point scale. In Study 1, we obtain the sentiment for all subphrases involved to ensure that subphrases and their controls have the same sentiment. We then discard phrases for which $A$ and/or $B$ do not have more than three controls each, where we restrict the controls to those whose sentiment is at most 1 point removed from the sentiment of $A$ (for $A'_n$) or $B$ (for $B'_n$).

In Study 2, we collect sentiment labels for all subphrase combinations, namely the remaining 259 phrases and the 1554 phrases in which a control subphrase is inserted. For a phrase "$A$ $B$", there are six controls: three that substitute $A$, and three that substitute $B$. Those substitutions could lead to ungrammatical constructions in spite of the data selection procedure, and participants can indicate that with a checkbox. Figure 4 in Appendix B displays example questions as shown to the participants. Participants were recruited via Prolific and annotated sentiment via a Qualtrics survey. In Study 1, 57 participants annotate 93 or 94 subphrases each. In Study 2, 90 participants annotate 60 or 61 subphrase combinations each. That way, every unique phrase and subphrase receives three annotations

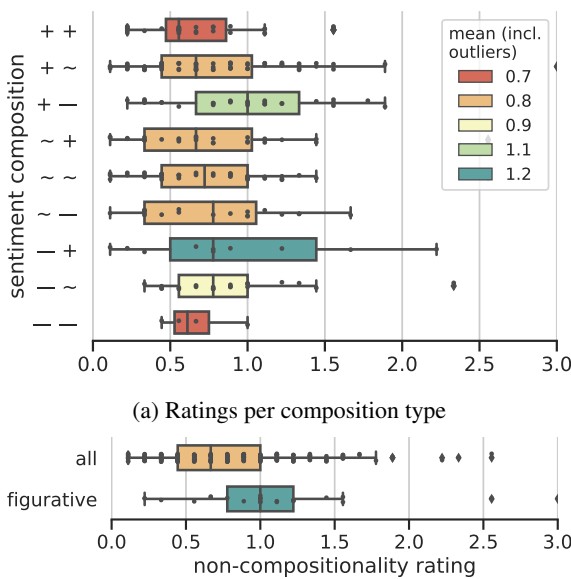

(a) Ratings per composition type

(b) Ratings for all and figurative phrases

Figure 3: MAXABS non-compositionality ratings a) per composition type ('-', '∼' and '+' refer to negative, neutral and positive), and b) for figurative examples.

total. The inter-annotator agreement rates obtained were 0.60 and 0.64 for Study 1 and 2, respectively, in terms of Krippendorff's $\alpha$ for ordinal data. Appendix B further discusses these studies along with ethical considerations and annotation statistics.

### 3.3 Computing non-compositionality ratings

We obtain one sentiment label per phrase by averaging the annotations from Study 2. Afterwards, the non-compositionality ratings for a phrase "$A$ $B$" are computed separately for $A$ and $B$. The rating for $A$ is the difference between the sentiment of "$A$ $B$" and the mean sentiment of phrases "$A'_n$ $B$" ($n \in \{1, 2, 3\}$), and vice versa for $B$. Together, the two ratings express the non-compositionality of "$A$ $B$". We compute four variants of the ratings: ALL, ALLABS, MAX, MAXABS, ALLCLEAN. The first two include $A$ and $B$ separately, the second two use one rating per phrase (the largest of the two). ALLCLEAN includes $A$ and $B$ separately but excludes any phrases that are considered ungrammatical (109 out of 1813 phrases involved in Study 2 were flagged for that).

### 4 Analysis of the ratings

What patterns can we identify in these ratings? We examine sentiment composition types, phrase lengths and syntactic categories of subphrases in Appendix C; only the sentiment composition type

| subphrase A | subphrase B | Rating | Sentiment | |
|---|---|---|---|---|
| | | | HUMAN | ROBERTA |
| a nearly terminal case | of the cutes | 4.11 | 5.33 | 2.00 |
| the franchise's best years | are long past | -4.11 | 0.33 | 1.00 |
| all the excitement | of eating oatmeal | -3.00 | 2.00 | 2.33 |
| a pressure cooker | of horrified awe | -2.56 | 1.00 | 3.67 |
| fans of the animated wildlife adventure show | will be in warthog heaven | 1.56 | 5.67 | 5.00 |
| a real human soul | buried beneath a spellbinding serpent's smirk | 1.56 | 4.67 | 5.00 |
| everyone involved with moviemaking | is a con artist and a liar | -1.33 | 0.00 | 1.00 |
| the modern master of the chase sequence | returns with a chase to end all chases | 1.11 | 6.00 | 5.00 |

Table 1: Examples of non-compositional phrases, with their MAX rating (§3.3 describes how these ratings are computed) and the sentiment assigned by the annotators and by ROBERTA-LARGE (details on how the model was trained are contained in §5). Red indicates negative sentiment, green indicates positive sentiment.

displays a clear pattern, illustrated by the ratings' distributions in Figure 3. The most compositional are the cases where the subphrases share their positive/negative sentiment, whereas combining opposites is the least compositional. Most phrases have an absolute rating within 1 point of our 7-point scale; only for 67 out of the 259 phrases, the MAXABS non-compositionality rating exceeds 1.

What characterises the least compositional examples?[2] The most prominent pattern is that figurative language is over-represented, which we can quantitatively illustrate by annotating all phrases as figurative and literal; the resulting MAXABS distributions differ substantially (Figure 3). In Table 1, some examples of **figures of speech** are the "pressure cooker" (a container metaphor implying a stressful situation, Kövecses and Kövecses, 1990), the "nearly terminal case of the cutes" (suggesting one can die of cuteness for emphasis), the "serpent's smirk" (metaphorically used to invoke connotations about evil) and the hyperbole of "everyone [...] is a con artist and a liar".

Other atypical sentiment patterns that we observe require **common-sense reasoning** about terms that act as contextual valence shifters (Polanyi and Zaenen, 2006), e.g. to understand that "are long past" implies something about *current* times, or that "eating oatmeal" relates to the blandness of that experience. Lastly, we also observe **discourse relations** between subphrases that modify the sentiment in a non-compositional manner. In "fans of the animated wildlife adventure show will be in warthog heaven", the parallel between 'wildlife' and 'warthog heaven' amplifies the positive sentiment in a way that would not have happened for fans of a fashion show. Similarly,

"returns with a chase to end all chases" functions differently when it concerns the "master of the chase sequence" rather than anyone else. These examples illustrate sentiment compositions are often nuanced and that there is a long tail of atypical non-compositional phenomena.

## 5 Evaluating sentiment models

How can we employ the non-compositionality ratings to better understand the quality of sentiment systems? We illustrate this by recreating the ratings using state-of-the-art pretrained neural models and comparing them to the humans' ratings.

**Experimental setup** To obtain the ratings from models, we adapt SST[3] to use the 7-point scale and exclude the phrases of interest from the training data. Per model type, we fine-tune three model seeds that we evaluate on the SST test set using $F_1$-score, and on NONCOMPSST using a) the correlation of the models' and humans' non-compositionality ratings (Pearson's $r$), and b) the $F_1$-score of NONCOMPSST phrases, using the humans' sentiment scores as labels. To obtain models' non-compositionality ratings, we average sentiment predictions from the three model seeds and apply the same postprocessing as applied to the human-annotated data (see §3.3).

We evaluate ROBERTA-BASE and -LARGE (Liu et al., 2019) along with variants of those models that are further trained on sentiment-laden data: TIMELM (the BASE model pretrained on tweets by Loureiro et al., 2022); the model of Camacho-collados et al. (2022), which is TIMELM fine-tuned to predict tweets' sentiment; BERTTWEET (a BASE

---

[2]We include them in Appendix C, in Table 3.

[3]The authors published the unprocessed original annotations by Amazon Mechanical Turk annotators of Socher et al. (2013); we process this data into SST-7.

| Model name | SST | NonCompSST | | | | | | |
|---|---|---|---|---|---|---|---|---|
| | $F_1$ | Max
$r$ | MaxAbs
$r$ | All
$r$ | AllAbs
$r$ | AllClean
$r$ | All
$F_1$ | Top 67
$F_1$ |
| *- Pretrained* | | | | | | | | |
| Roberta-base, Liu et al. | .43 | .36 | .31 | .41 | .22 | .43 | .40 | .32 |
| Roberta-large, Liu et al. | .47 | .42 | .38 | .44 | .30 | .46 | .47 | .37 |
| *- Pretrained using sentiment-laden data* | | | | | | | | |
| TimeLM, Loureiro et al.[B] | .43 | .30 | .32 | .34 | .25 | .36 | .43 | .38 |
| BertTweet, Nguyen et al.[B] | .46 | .22 | .20 | .25 | .15 | .27 | .43 | .30 |
| *- Finetuned using sentiment-laden data* | | | | | | | | |
| Camacho-collados et al.[B] | .45 | .33 | .36 | .36 | .22 | .38 | .49 | .43 |
| Pérez et al.[B] | .44 | .13 | .21 | .20 | .16 | .21 | .42 | .26 |
| Hartmann et al.[L] | .46 | .38 | .34 | .41 | .28 | .44 | .45 | .33 |
| IMDB Roberta[B] | .44 | .37 | .31 | .41 | .24 | .44 | .47 | .43 |

Table 2: Model evaluation using SST ($F_1$) and NonCompSST, according to correlation (Pearson's $r$) between the humans' and the models' non-compositionality ratings, and the $F_1$ of the 259 phrases and the 67 most non-compositional ones, measured using the humans' annotations as labels. We indicate whether models are BASE ($B$) or LARGE ($L$) and underline the highest performance per column.

model pretrained on tweets by Nguyen et al., 2020); the model of Pérez et al. (2021) (BertTweet fine-tuned to predict tweets' sentiment); Roberta-large (Hartmann et al., 2021) fine-tuned on sentiment of comments from social media posts; and finally Roberta-base fine-tuned on sentiment from IMDB movie reviews (Maas et al., 2011).[4]

**Results** The results in Table 2 suggest that even though the systems have very similar performance on the SST test set, there are differences in terms of the NonCompSST ratings: Roberta-base has the lowest SST $F_1$, but is the second-best BASE model in terms of $r$, only outperformed by IMDB-Roberta (i.e. the variant fine-tuned on movie reviews, the domain of SST). The Roberta-large model outperforms both Hartmann et al.'s model and the BASE models in terms of the SST $F_1$ and NonCompSST correlations. Together, these observations suggest that pretraining or fine-tuning using data from a different domain can harm models' ability to capture nuanced sentiment differences required to estimate NonCompSST ratings. The SST performance is less sensitive to this, suggesting that our resource can provide a complementary view of sentiment systems.

Finally, we also inspect the NonCompSST $F_1$-scores for all 259 phrases and the 67 phrases with the highest non-compositionality ratings according to the human annotators, for which results are included in the final two columns of Table 2. On that subset, IMDB-Roberta and the model of

Camacho-collados et al. (2022) achieve the highest $F_1$-score. These scores emphasise that for the top 67, sentiment is substantially harder to predict: non-compositional examples indeed present a larger challenge to sentiment models than compositional examples do.

## 6  Conclusion

Sentiment and compositionality go hand-in-hand: success in sentiment analysis is often attributed to models' capability to 'compose' sentiment. Indeed, the sentiment of a phrase is reasonably predictable from its subphrases' sentiment, but there are exceptions due to the ambiguity, contextuality and creativity of language. We made this explicit through an experimental design that determines *non*-compositionality ratings using humans' sentiment annotations and obtained ratings for 259 phrases (§3). Even though most phrases are fairly compositional, we found intriguing exceptions (§4), and have shown how the resource can be used for model evaluation (§5). For future sentiment analysis approaches, we recommend a multi-faceted evaluation setup: to grasp the nuances of sentiment, one needs more than compositionality.

## Limitations

Our work makes several limiting assumptions about compositionality in the context of sentiment:

1. We maintain a simplistic interpretation of the composition 'function' but are aware that compositionality is considered **vacuous** by some (Zadrozny, 1994) since by using a generic

---

[4]See Appendix D for further details on the experimental setup used to fine-tune these models on SST. Visit our repository for the data and code.

notion of a 'function', any sentiment computation can be considered compositional. We, therefore, only consider some phrases non-compositional because of the strict interpretation of that 'function'. As a result, one might argue that whether a phrase such as "all the excitement of eating oatmeal" is non-compositional in terms of its sentiment is debatable. We agree with that; if you represent the sentiment with a very expressive representation, every sentiment computation is compositional. Our results only apply *given* a very narrow interpretation of compositionality.

2. In this work, we restrict the notion of **meaning compositions** to the notion of **sentiment compositions**. Hence, there might be phrases that behave compositionally in terms of sentiment but are considered non-compositional otherwise. For instance, "rotten apple" carries negative sentiment, both literally and figuratively, and might thus be considered compositional in terms of sentiment.

In addition to that, the resource we developed has technical limitations:

1. Human annotators can provide **unreliable sentiment annotations**: they do not necessarily agree with one another, may lose focus while performing the task or may misunderstand the linguistic utterances they annotate. As a result, the resource inevitably contains some sentiment ratings that are inaccurate.

2. The resource we develop is small in **size**, which limits the robustness of the results when using the resource for experimentation. We would like to point out, however, that ≫259 annotations were involved in obtaining the ratings for these 259 phrases. The results illustrate that, in spite of these limitations, the resource can still lead to valuable conclusions.

Finally, the evaluation of the models in §5 is somewhat limited, considering that the various models have been fine-tuned or pretrained by the mentioned authors using different experimental setups. Even though we then apply the same setup to fine-tune on SST, these differences need to be kept in mind when interpreting the results.

## Acknowledgements

We thank Kenny Smith and Ivan Titov for their suggestions throughout this project and Matthias Lindemann for his comments on a draft of this article. VD is supported by the UKRI Centre for Doctoral Training in Natural Language Processing, funded by the UKRI (grant EP/S022481/1) and the University of Edinburgh, School of Informatics and School of Philosophy, Psychology & Language Sciences.

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

## A  Materials

We require data for which to obtain the non-compositionality ratings and the control stimuli used to construct the default mapping (step 1 in Figure 2).

**Data selection**  We obtain our data from the Stanford Sentiment Treebank (SST), a resource containing 11,855 sentences from movie reviews collected by Pang and Lee (2005), and annotated with sentiment labels by Socher et al. (2013). The sentences were parsed with the Stanford Parser (Klein and Manning, 2003) to allow for sentiment annotations of phrases in addition to full sentences. The dataset includes sentiment labels for all nodes of those parse trees. The sentiment labels were obtained from Amazon Mechanical Turk annotators, who indicated the sentiment using a multi-stop slider bar with 25 ticks.

We select potential phrases to include in our dataset, by applying the following constraints to the SST constituents. 1) They are marked as constituents by a state-of-the-art constituency parser, and have two subphrases.[5] This ensures that the segmentation of the two parts is somewhat reasonable, and excludes compositions of three or more constituents. 2) The constituents' subphrases have 3-8 tokens, excluding punctuation. We exclude longer constituents, cognisant of the higher cognitive load that labelling those phrases would have for our participants. 3) The constituents do not contain a named entity, such as a movie star or the name of a film. Not all participants may know these names and the associated sentiment, hence we opt for excluding them. 4) The standard deviation of the original SST annotations lies within a range of five (on the original 25-point scale), to exclude the most ambiguous phrases. That standard deviation is computed over the three annotations per phrase obtained by Socher et al. (2013).

**Selection of control subphrases**  How do we select the control stimuli? They should have the same polarity as the part they are replacing. In that way, we can obtain the default mapping without altering the "polarities of the parts" (Moilanen and Pulman, 2007). Each phrase from the data selection step has two subphrases. We distribute those subphrases into groups based on their SST sentiment (on an 11-point scale, from 0.0 to 1.0) and phrase type (NP, VP, PP, SBAR).

For each phrase – consisting of subphrases $A$ and $B$ – we randomly select 32 candidate control subphrases from those groups and manually narrow those 32 down to 8 control subphrases (4 for $A$, 4 for $B$). Where needed, small modifications are made to the control stimuli to create grammatical agreement between $A'_n$ and $B$, and $A$ and $B'_n$. During the manual annotation, examples for which fewer than 8 suitable control stimuli remain are removed until 500 phrases remain to be used in the annotation study.

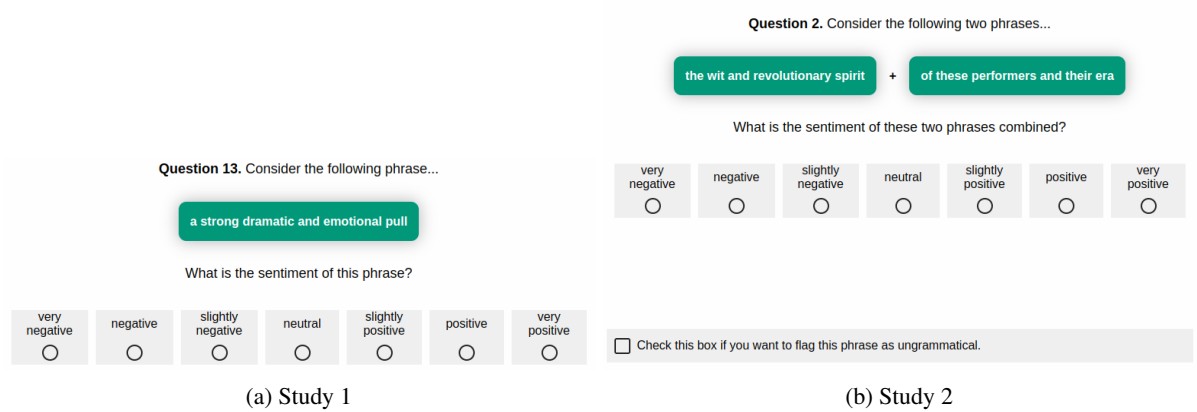

(a) Study 1                                        (b) Study 2

Figure 4: Illustration of the question formatting in Study 1 and 2.

## B  Studies

The data annotation studies have been approved by the local ethics committee of the institute to which the authors belong. Prior to starting the questionnaire, the participants are presented with a *Participant*

---

[5]We use stanza's tokenizer, parser and named entity recognition model for English (Qi et al., 2020), https://stanfordnlp.github.io/stanza.

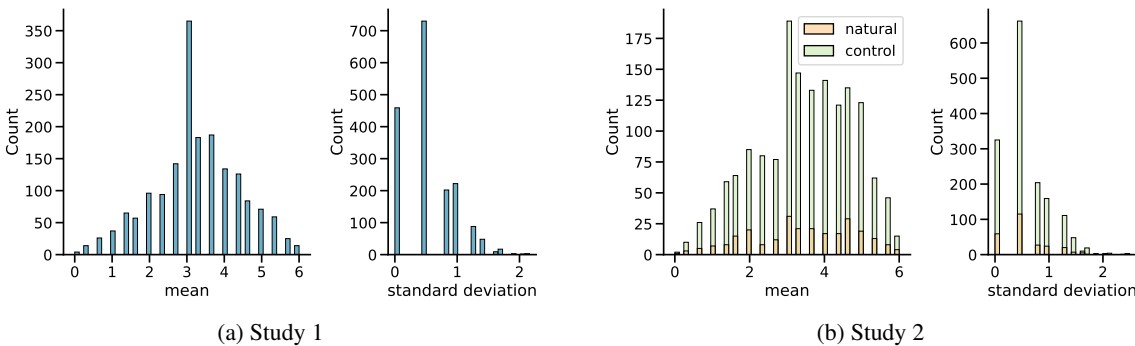

|  (a) Study 1 | (b) Study 2 |

Figure 5: Distributions of the means and standard deviations observed for Studies 1 and 2, computed over the three annotations obtained per stimulus.

*Information Sheet* that explains their data protection rights, the goal of the study and the compensation that they will receive. We set out to pay participants at a rate of £11 per hour (above the minimum wage), and estimated both studies to take 12 minutes to fill out.

**Procedure Study 1** Participants annotate the data via a survey on the Qualtrics platform. We first ask participants to familiarise themselves with the sentiment labels, by showing seven subphrases from the SST dataset and requesting the participants to match them with the correct sentiment label. Afterwards, the correct answers are shown. These subphrases and their labels are taken from the SST dataset. Afterwards, the participants are shown 93 or 94 subphrases in isolation, and annotate them (the number of stimuli could not evenly be divided over participants while adhering to the desired study length of approximately 12 minutes). At the very end, the participants can provide feedback on the task. This study constitutes step 2 in Figure 2.

**Procedure Study 2** The second study has a very similar procedure. First, the participants familiarise themselves with matching subphrases with the seven labels. Afterwards, they are asked to use the same labels but now assign them to subphrase combinations. In both cases, the correct answers are shown, afterwards. We also explain that the subphrase combinations can seem odd semantically but that if the participants encounter *ungrammatical* subphrase combinations, they can flag that using a checkbox. Afterwards, the participants are shown 60 or 61 subphrase combinations that they then annotate. We clearly mark the segmentation of the phrase into two parts using colour. Participants receive fewer questions compared to the first study, due to the longer explanation that is included in Study 2, and due to the fact that annotating longer phrases may simply take longer. At the very end, the participants can provide feedback on the task. This study constitutes step 3 in Figure 2.

**Participants** We recruited participants via Prolific, requiring them to be located in the United Kingdom. Further selection criteria were that they have listed English as their first language and that they have a perfect Prolific approval rate over a minimum of 20 completed studies. Participants were initially compensated £2.20 for both studies. The median completion time for Study 1 was 10:30 minutes. The median completion time for Study 2 was 13:16 minutes; these participants were paid a bonus to increase the mean reward per hour to £11. In both studies, we excluded the results for participants whose sentiment annotations for the practice questions were, on average, more than 1 point removed from the correct labels (according to SST), or for whom the correlation between their responses and the labels of these practice questions was below 0.8 (according to Spearman's $\rho$).

**Annotations** Figure 5 includes the distributions over the sentiment labels that the participants assign. For 104 phrases, there was one annotator that indicated that the phrase was ungrammatical, and for 5 phrases, two annotators indicate that the phrase was ungrammatical. Sentiment label '3' represents the neutral label. In Study 1, this label is much more frequent compared to Study 2. This is in line with findings from Socher et al. (2013), who established that the longer the phrase, the more frequent the phrase is considered sentiment-laden. Across the board, the positive phrases are slightly over-represented. In Study 2, there are both natural and control stimuli, but the distributions are not substantially different.

# C Visualisation of the non-compositionality ratings

In Figure 6 and Figure 7 we separate the MAXABS and ABS non-compositionality ratings per sentiment composition type, per length combination, and per syntactic category combination. Only sentiment composition type is a clear dominant factor leading to higher ratings. Even though there are some longer phrase combinations with high average ratings, data from those categories is also rather scarce.

Table 3 gives 60 examples of phrases whose MAXABS rating exceeds 1, ranked from the highest to the lowest.

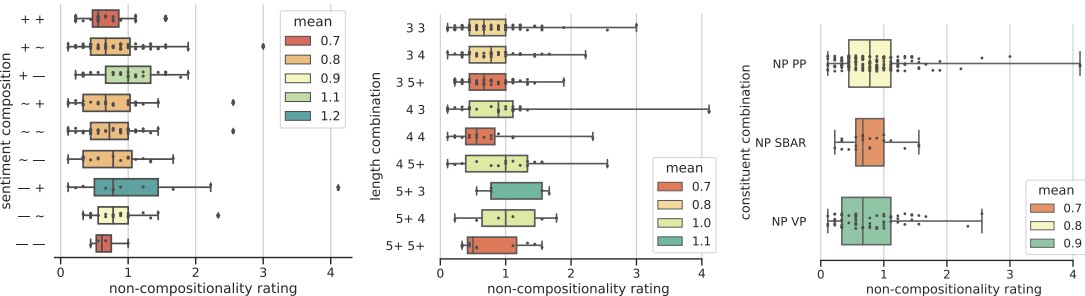

Figure 6: MAXABS ratings (that assign phrase "*A B*" the highest rating out of the two obtained for *A* and *B*), shown separately per sentiment composition type, length combination type and syntactic category.

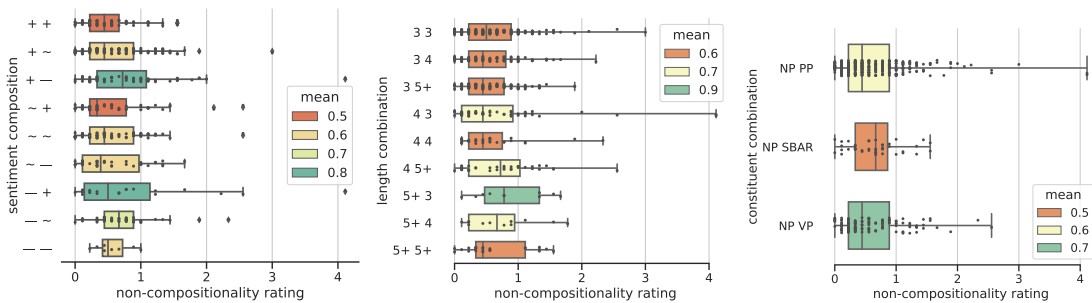

Figure 7: ABS ratings (that include the ratings for subphrases *A* and *B* separately), shown per sentiment composition type, length combination type and syntactic category.

| Fig. | Phrase A | Phrase B | Rating | Sent. H | Sent. R-L |
|---|---|---|---|---|---|
| ✓ | a nearly terminal case | of the cutes | 4.11 | 5.33 | 2.00 |
|  | the franchise's best years | are long past | -4.11 | 0.33 | 1.00 |
| ✓ | all the excitement | of eating oatmeal | -3.00 | 2.00 | 2.33 |
|  | just a simple fable | done in an artless style | -2.56 | 1.00 | 1.33 |
| ✓ | a pressure cooker | of horrified awe | -2.56 | 1.00 | 3.67 |
|  | something creepy and vague | is in the works | 2.33 | 3.67 | 2.00 |
|  | a no-surprise series | of explosions and violence | -2.22 | 0.67 | 2.00 |
|  | the numerous scenes | of gory mayhem | -1.89 | 1.67 | 3.00 |
|  | the momentary joys | of pretty and weightless intellectual entertainment | -1.89 | 2.00 | 4.00 |
|  | flourishes – artsy fantasy sequences – | that simply feel wrong | -1.78 | 1.00 | 1.67 |
|  | the overall feel of the film | is pretty cheesy | -1.67 | 0.67 | 1.00 |
|  | the contrived nature | of its provocative conclusion | -1.67 | 1.33 | 2.00 |
| ✓ | fans of the animated wildlife adventure show | will be in warthog heaven | 1.56 | 5.67 | 5.00 |
|  | a pointed little chiller | about the frightening seductiveness of new technology | -1.56 | 2.67 | 4.00 |
| ✓ | a real human soul | buried beneath a spellbinding serpent's smirk | 1.56 | 4.67 | 5.00 |
|  | the comic heights | it obviously desired | -1.56 | 2.67 | 3.67 |
|  | the belly laughs | of lowbrow comedy | 1.56 | 4.67 | 4.00 |
|  | the intellectual and emotional pedigree | of your date | -1.56 | 2.67 | 3.33 |
|  | the experience of going to a film festival | is a rewarding one | 1.56 | 6.00 | 5.00 |
|  | a serious exploration | of nuclear terrorism | 1.56 | 4.33 | 4.00 |
|  | a simple tale | of an unlikely friendship | 1.56 | 5.33 | 4.00 |
|  | the real charm | of this trifle | -1.44 | 3.67 | 4.00 |
|  | the human story | is pushed to one side | -1.44 | 2.00 | 2.00 |
|  | a pale imitation | of the real deal | -1.44 | 0.67 | 1.00 |
|  | a main character | who sometimes defies sympathy | -1.44 | 1.67 | 2.33 |
| ✓ | the obligatory moments | of sentimental ooze | -1.44 | 2.00 | 3.00 |
|  | the actresses in the lead roles | are all more than competent | 1.44 | 5.33 | 5.00 |
|  | the debate it joins | is a necessary and timely one | 1.44 | 4.67 | 5.00 |
|  | an audacious tour | of the past | -1.33 | 2.67 | 3.67 |
|  | only a document | of the worst possibilities of mankind | 1.33 | 2.00 | 1.00 |
|  | the life experiences | of a particular theatrical family | 1.33 | 4.67 | 3.00 |
|  | the entire point | of a shaggy dog story | 1.33 | 4.00 | 3.00 |
|  | an inexpressible and drab wannabe | looking for that exact niche | 1.33 | 2.00 | 1.00 |
|  | everyone involved with moviemaking | is a con artist and a liar | -1.33 | 0.00 | 1.00 |
|  | a very original artist | in his medium | 1.33 | 5.33 | 5.00 |
|  | the visuals and eccentricities | of many of the characters | 1.33 | 4.33 | 3.00 |
| ✓ | a violent initiation rite | for the audience | -1.22 | 2.00 | 2.67 |
|  | these curious owners | of architectural oddities | -1.22 | 2.67 | 3.00 |
|  | this engaging mix | of love and bloodletting | 1.22 | 5.00 | 4.67 |
|  | a sudden lunch rush | at the diner | -1.22 | 2.67 | 3.00 |
|  | the true potential | of the medium | -1.22 | 3.00 | 4.00 |
| ✓ | a therapeutic zap | of shock treatment | -1.22 | 3.00 | 3.33 |
|  | the exotic world | of belly dancing | -1.22 | 3.00 | 3.67 |
| ✓ | the real story | starts just around the corner | 1.22 | 3.67 | 3.00 |
|  | the most offensive thing | about the movie | -1.22 | 1.00 | 1.00 |
|  | the issue of faith | is not explored very deeply | -1.22 | 1.00 | 2.00 |
|  | a big box | of consolation candy | 1.22 | 3.67 | 3.00 |
|  | the playful paranoia | of the film's past | 1.22 | 4.00 | 3.67 |
|  | the overlooked pitfalls | of such an endeavour | -1.22 | 2.00 | 3.00 |
| ✓ | the modern master of the chase sequence | returns with a chase to end all chases | 1.11 | 6.00 | 5.00 |
|  | the sheer beauty | of his images | 1.11 | 5.67 | 5.00 |
|  | a compelling slice | of awkward emotions | 1.11 | 4.67 | 5.00 |
|  | a great actress | tearing into a landmark role | -1.11 | 4.67 | 5.33 |
|  | much of the writing | is genuinely witty | 1.11 | 5.00 | 5.00 |
|  | an especially poignant portrait | of her friendship | -1.11 | 3.67 | 5.00 |
|  | the insight and honesty | of this disarming indie | -1.11 | 3.67 | 5.00 |
|  | the emotional arc | of its raw blues soundtrack | 1.11 | 4.33 | 3.67 |
|  | the kind of art shots | that fill gallery shows | 1.11 | 4.67 | 4.00 |
|  | a new software program | spit out the screenplay | 1.11 | 2.67 | 3.00 |
|  | a very good time | at the cinema | 1.11 | 5.67 | 5.67 |

Table 3: 60 phrases with a rating that exceeds an absolute value of 1. We indicate the rating from the MAX variant of our resource. We indicate whether we believe the phrase to represent a figurative phrase, and also provide the sentiment assigned by the annotators (**Sent. H.**) and by ROBERTA-LARGE (**Sent. R-L**)

## D Models & model evaluation

**Experimental setup**   We fine-tune the following models, all obtained from the HuggingFace model hub:

- ROBERTA-BASE https://huggingface.co/roberta-base
- ROBERTA-LARGE https://huggingface.co/roberta-large
- TIMELM (Loureiro et al., 2022):
  https://huggingface.co/cardiffnlp/twitter-roberta-base
- BERTTWEET (Nguyen et al., 2020): https://huggingface.co/vinai/bertweet-base
- Camacho-collados et al. (2022):
  https://huggingface.co/cardiffnlp/twitter-roberta-base-sentiment-latest
- Pérez et al. (2021):
  https://huggingface.co/finiteautomata/bertweet-base-sentiment-analysis
- Hartmann et al. (2021):
  https://huggingface.co/j-hartmann/sentiment-roberta-large-english-3-classes
- IMDB ROBERTA-BASE: https://huggingface.co/textattack/roberta-base-imdb

The experimental setup used to fine-tune them on SST is as follows: we use a batch size of 32, learning rate $5e{-}6$, the Adam optimiser with a cosine-based warmup scheduler (warmup is 20%). We train for 5 epochs, selecting the best model based on the validation data and evaluate that on the test sets. We do not experiment with these hyperparameters extensively, as they are within the range of recommended settings and all models use the same base models (ROBERTA-BASE or ROBERTA-LARGE).

We train using NVIDIA A100-SXM-80GB GPUs on which training one model for one seed takes up to 30 minutes for the BASE model, and 50 minutes for the LARGE model.

**Characterising our experiments**   Through our experiments, we contribute to generalisation evaluation in NLP: evaluating models' generalisation capabilities beyond standard i.i.d. testing, in particular through the evaluation on the 67 most non-compositional phrases. We characterise our experiments using the taxonomy of Hupkes et al. (2023): our experiments are cognitively motivated and focus on (non-)compositional generalisation. By separating inputs based on compositionality, we create a covariate shift between train and test data. That shift has a partitioned natural source, since the stimuli are from a human-written corpus, but the split we create is curated. In our experiments, we examine the influence of fine-tuning on SST-7 and evaluating on non-compositional examples (fine-tune train-test locus) but also examine the effect of a data shift in the pretrain-train locus, by considering models with different pretraining corpora.

| Motivation | | | |
|---|---|---|---|
| *Practical* | *Cognitive* ☐ | *Intrinsic* | *Fairness* |
| **Generalisation type** | | | |
| *Compositional* ☐ | *Structural* | *Cross Task*   *Cross Language*   *Cross Domain* | *Robustness* |
| **Shift type** | | | |
| *Covariate* ☐ | *Label* | *Full* | *Assumed* |
| **Shift source** | | | |
| *Naturally occuring* | *Partitioned natural* ☐ | *Generated shift* | *Fully generated* |
| **Shift locus** | | | |
| *Train–test* | *Finetune train–test* ☐ | *Pretrain–train* ☐ | *Pretrain–test* |