# OpenReview forum: "Non-Compositionality in Sentiment: New Data and Analyses"
_EMNLP/2023/Conference — EMNLP 2023 Findings_

### Official Review · Reviewer_btm9 · 2023-07-24

**Soundness:** 3

**Excitement:**

4: Strong: This paper deepens the understanding of some phenomenon or lowers the barriers to an existing research direction.

**Paper Topic And Main Contributions:**

- The paper aims to identify which phrases are compositional in terms of sentiment and which are not. Non-compositionality means that the sentiment of the entire phrase is "unexpected given the sentiment of the individual parts".
- The authors create a new datasource (subset of SST2) with non-compositionality labels. They analyze the datasource and run classification models on it.


**Questions For The Authors:**

- Q1: Appendix A, L540:
	- "4) The standard deviation ...": What does it mean that the std dev lies within a range of 5? What's the std dev computed over? Only the sentiment scores of tokens in 1 phrase? Over all tokens in the dataset?
	- Assuming a normal distribution of sentiment values, a value lying outside of 5 std dev is very unlikely to occur in the dataset (probability of about 5.7e-7). How many phrases does that apply to?
- Q2: Fig 3
	- What are the number of examples in each "sentiment composition" group?
	- The differences in mean values are quite small. Have you run a significance test showing if they are indeed statistically significant?
	- How do you explain the examples that are "- +" that have a very low non-compositionality rating? Are these examples mentioned in the introduction (e.g., "yeah right")? Adding such examples for illustration purposes would be useful.
- Q4: L182: "Together, the two ratings...": How are the 2 numbers combined into one non-compositionality scores?
- Q5: L184: Same question for ALL, ALLABS and ALLCLEAN. Correlation in Sec. 5/Tab. 1 indicates they are merged into 1 number. How?
- Q6: Sec. 5
	- It would be useful to know the number of training examples and how they were selected. Is it all of SST training data minus the overlapping examples?
	- It would be very interesting to see the model performance being broken down by category as in Fig. 3 to further understand strengths and weaknesses of the models.
- Q7: Tab. 1: It's expected that the *ABS variants perform worse than their non-*ABS counterparts, because the absolute value is correlated with the linear 7-point Likert scale of SST. A negative value in the former (which would mean low value in Likert scale) is mapped to a positive value (which would mean a higher value in Likert scale). As such seeing these 2 variants in a correlation analysis does not provide any useful information.


**Reasons To Accept:**

- The analysis is interesting and gains some insights into compositionality of sentiment labels.
- The created dataset is a nice resource.
- The paper is nicely written and most steps seem very well thought through.


**Reasons To Reject:**

- In some places details are missing (see questions below).
- It's not clear to me how strong "sentiment composition" types relate to non-compositionality (Fig. 3). The absolute differences in the figure are rather small and details are missing in order to judge the strength of the statements (e.g., number of examples per box, test for significance).

**Reproducibility:**

3: Could reproduce the results with some difficulty. The settings of parameters are underspecified or subjectively determined; the training/evaluation data are not widely available.

**Reviewer Confidence:**

4: Quite sure. I tried to check the important points carefully. It's unlikely, though conceivable, that I missed something that should affect my ratings.

**Typos Grammar Style And Presentation Improvements:**

- Be careful with shrinking vertical spaces, e.g., under table/figure captions. It blends into the text too much.
- The paper could be easier to follow if it was using "compositionality" vs "non-compositionality", because the paper often talks about low non-compositionality (which is basically double negation).

---

> ### Author Rebuttal · Authors · 2023-08-28
>
> Many thanks for the elaborate review! These questions are very useful to further improve the writing of the paper. We answer your questions below.
>
> **Question 1**: The Stanford Sentiment Treebank originally collected sentiment labels using a multi-stop slider bar with 25 ticks. Every phrase was annotated by three annotators. The “range of five” refers to the standard deviation among those three annotations. This excludes approximately 5% of the data and acts as a filter that removes phrases that were very ambiguous in the original data annotation.
>
> **Question 2**:
> - Individual data points are indicated on the graph with dots. The group sizes vary from 4 to 68, with a mean of 28. Given the small dataset size and even smaller subgroup sizes, it is hard to draw definite conclusions about the significance of results, but within the results displayed in Figure 3, the `+-’ non-compositionality ratings are significantly (p<0.05) greater than those of six other composition types (~~, +~, ~+, ~-, ++, --) as per Welch’s t-test (because of the unequal variance and varying group sizes). The figurative expressions also have a significantly greater mean compared to the controls (p=0.013).
> - Regarding the examples with low non-compositionality scores and `- +’ sentiment: opposing sentiment is not necessarily a reason for non-compositional sentiment, even if the results show that it is more likely. In “A near terminal case of the cutes” (high non-compositionality and also - +) it is the case that the contrast amplifies in a non-compositional way, but in some other cases, such as “the problems and characters it reveals are universal and involving” (non-compositionality score of 0.22 for MaxABS) it is not. “the problems and characters it reveals <insert positive phrase>”/"<insert negative phrase> are universal and involving" thus behave more predictably, compositionally, compared to “a near terminal case of <insert positive phrase>”/"<insert negative phrase> of the cutes".
>
> **Question 3**: you may have omitted Q3 from your review. Feel free to post it in the comments during the discussion period if this was by accident.
>
> **Questions 4 and 5** (how are the two ratings combined into one number?): That depends on the ranking: in AllABS, All and AllClean, they are not combined into one number; they are included as two separate ratings. In Max and MaxABS, they are combined by taking the maximum of the two.
> Therefore, the correlations in Table 1 / Section 5 are computed over 259 x 2 numbers for All, AllClean and AllABS.
>
> **Question 6**: the training set includes the phrases normally included in training, excluding sentences that contain any of the phrases involved in obtaining the ranking (note that these include simple phrases such as “of a movie” that might appear in multiple sentences from the dataset). The SST training data normally includes 320k phrases (including sentences and subphrases from sentences), of which 160k unique. We train on 274k phrases.
>
> Regarding the breakdown of performance: when breaking down by composition type, some correlations are not reliable due to small subgroup sizes (Pearson’s r with p-values >> 0.05). As an alternative, two pieces of information are provided below:
> - The first markdown table shows the F1 of the 259 phrases, using the humans’ predictions as labels (different from the F1 for the test set from Table 1), and contrasts that to the F1 computed over just the 67 least compositional phrases.
> - The second markdown table shows the MAE for the MaxABS ranking, comparing the non-compositionality scores of the models to that of humans per composition type.
>
> Both results illustrate that the composition types with higher non-compositionality ratings are also more challenging in terms of predicting the sentiment labels. Thanks for asking for these additional results; we would like to expand upon the results discussion in the camera-ready version of the paper.
>
> | model | F1 over all 259 phrases | F1 over 67 least compositional |
> | ----- | ----- |  ----- |
> | roberta-base | 40 | 32 |
> | roberta-large | 47 | 37 |
> | timelm | 43 | 38 |
> | berttweet | 43 | 30 |
> | camacho-collados et al | 49 | 43 |
> | perez et al | 42 | 26 |
> | hartmann et al | 45 | 33 |
> | imdb roberta | 47 | 43 |
>
>
> | model | ~ ~ | + ~ | — ~ | ~ + | ~ — | — + | + + | + — | — — |
> | ---- | ---- | ---- | ---- | ---- | ---- | ----| ----| ----| ----
> | roberta-base | 0.40| 0.49| 0.58| 0.50| 0.47| 0.92| 0.40| 0.49| 0.56 |
> | roberta-large | 0.43| 0.43| 0.57| 0.44| 0.44| 0.99| 0.36| 0.50| 0.61 |
> | timelm | 0.43| 0.50| 0.58| 0.47| 0.47| 0.75| 0.40| 0.50| 0.39 |
> | berttweet | 0.44| 0.47| 0.55| 0.54| 0.47| 0.77| 0.34| 0.67| 0.56 |
> | camacho-collados et al | 0.38| 0.45| 0.56| 0.49| 0.39| 0.63| 0.38| 0.49| 0.42 |
> | perez et al | 0.48| 0.55| 0.53| 0.56| 0.50| 0.73| 0.35| 0.65| 0.56 |
> | hartmann et al | 0.43| 0.44| 0.53| 0.49| 0.51| 0.99| 0.35| 0.48| 0.36 |
> | imdb roberta | 0.34| 0.47| 0.55| 0.51| 0.56| 0.90| 0.36| 0.51| 0.56 |
> | *mean* | 0.42 | 0.48 | 0.56 | 0.50 | 0.48 | 0.84 | 0.37 | 0.54 | 0.50 |
>
>
> **Question 7**: You are correct, and we mainly included all performance measures for consistency; the non-ABS variants might be useful during model analysis since they provide a more accurate representation of the targets, but the ABS variants are more intuitive during the data analysis. After all, we set out to compute a non-compositionality metric, which is more easily understood by the absolute value.
>
> Thanks for going to great lengths in your review, even pointing out potential representation improvements. We were indeed short on space – 4 pages is a challenging paper length – and will gladly use some space from the 5th page to increase figures and caption sizes. We will revise the wording of the non-compositionality ratings carefully but opted for the somewhat odd phrasing of "low/high non-compositionality rating" because the *non-compositionality* rating is the central theme and is depicted in Figure 3 / Table 2, where higher numbers represented higher non-compositionality ratings.

---

### Official Review · Reviewer_SRwS · 2023-08-02

**Soundness:** 3

**Excitement:**

3: Ambivalent: It has merits (e.g., it reports state-of-the-art results, the idea is nice), but there are key weaknesses (e.g., it describes incremental work), and it can significantly benefit from another round of revision. However, I won't object to accepting it if my co-reviewers champion it.

**Paper Topic And Main Contributions:**

The author argues that most of the existing sentiment analysis research is combinatorial. So he did research on non-combinatorial sentiment analysis.

contributions are as follows: a) a methodology for obtaining those non-compositionality ratings, b) a resource of ratings for 259 phrases along with an analysis of that resource, and c) an evaluation of computational models for sentiment analysis.


**Questions For The Authors:**

Can you summarize the conclusion of this paper?

**Reasons To Accept:**

This article presents a new perspective (combinatorial and non-combinatorial) to examine the prevailing sentiment analysis methods.

**Reasons To Reject:**

1. Although this article provides a fresh perspective, it does not reach a conclusion. Or rather, the article suggests that existing sentiment analysis models are already capable of handling both combinatorial and uncombinatorial sentences.
2. Only F1 and Pearson'R are used for evaluation, and MAE and MES will be more detailed when added.

**Reproducibility:**

5: Could easily reproduce the results.

**Reviewer Confidence:**

3: Pretty sure, but there's a chance I missed something. Although I have a good feel for this area in general, I did not carefully check the paper's details, e.g., the math, experimental design, or novelty.

---

> ### Author Rebuttal · Authors · 2023-08-28
>
> Thanks for your thoughts about our work! It’s good to see that you appreciate the new perspective on sentiment analysis that goes beyond the standard methods of evaluation.
>
> **Regarding your question about the conclusion of this paper**: let us rephrase the conclusion using new words. Our findings are four-fold:
> - Firstly, we have successfully established a methodology to obtain non-compositionality ratings from human annotators. This is not as simple as one might expect when thinking about how to obtain those ratings: meaning composition is a process that occurs implicitly, and we cannot simply ask “is this sentiment compositional”. Therefore, care went into designing our experimental protocols, which led to a successful ranking for the 259 phrases involved.
> - Secondly, we have confirmed what is generally believed, namely that sentiment is a largely compositional process, where the sentiment of a phrase is fairly predictable from the sentiment of the subphrases.
> - Thirdly, nonetheless, we confirm that a subset of the phrases is not actually compositional, and we focus on the 67 with the highest non-compositionality scores (>1), in particular. We observe that some composition types (such as + - and - +) are more likely to be perceived as non-compositional, and that figurative phrases are more prominent among non-compositional examples. Non-compositional examples are also more challenging to the models, which we elaborate on in the response to reviewer btm9, and we will elaborate on this in the paper.
> - Fourthly, we show that the non-compositionality ratings can be used for more fine-grained model evaluation in the task of sentiment analysis.
>
> Specifically, we would like to point out that you may be incorrect when mentioning that models can already handle compositional and non-compositional examples. In fact, models with reasonable SST F1 scores can have NonComp scores that are quite low, such as BertTweet and the model of Perez et al. in Table 1.
>
> **Regarding the metrics included in section 5**: we will elaborate the discussion of the results in the paper, including results presented below to reviewer btm9. In the table below, we report the MAE for the Pearson's correlations from Table 1. In general, the two metrics show similar patterns. We do believe the correlations to be a better fitting evaluation metric because of the emphasis on ranking examples based on their non-compositionality rather than getting the non-compositionality prediction exactly right.
>
> | model | Max (r / MAE) | MaxAbs (r / MAE) | All (r / MAE) | AllAbs (r / MAE) | AllClean (r / MAE) |
> | --- | --- | --- | --- | --- | --- |
> | roberta-base | 0.36   / 0.79  | 0.31   / 0.49  | 0.41   / 0.61  | 0.22   / 0.48  | 0.43   / 0.61  |
> | roberta-large | 0.42   / 0.74  | 0.38   / 0.47  | 0.44   / 0.59  | 0.30   / 0.46  | 0.46   / 0.58  |
> | timelm | 0.30   / 0.80  | 0.32   / 0.49  | 0.34   / 0.62  | 0.25   / 0.47  | 0.36   / 0.61  |
> | berttweet | 0.22   / 0.86  | 0.20   / 0.50  | 0.25   / 0.66  | 0.15   / 0.49  | 0.27   / 0.66  |
> | camacho-collados et al | 0.33   / 0.79  | 0.36   / 0.45  | 0.36   / 0.62  | 0.22   / 0.47  | 0.38   / 0.62  |
> | perez et al | 0.13   / 0.89  | 0.21   / 0.53  | 0.20   / 0.66  | 0.16   / 0.49  | 0.21   / 0.66 |
> | hartmann et al | 0.38   / 0.77  | 0.34   / 0.47  | 0.41   / 0.60  | 0.28   / 0.46  | 0.44   / 0.59 |
> | imdb roberta | 0.37   / 0.78  | 0.31   / 0.48  | 0.41   / 0.60  | 0.24   / 0.46  | 0.44   / 0.60 |

---

### Official Review · Reviewer_zamx · 2023-08-02

**Soundness:** 5

**Excitement:**

4: Strong: This paper deepens the understanding of some phenomenon or lowers the barriers to an existing research direction.

**Paper Topic And Main Contributions:**

The paper analyzes the role of non-compositionality of sentiment in phrases, understood as those whose sentiment deviates from what is expected. Given that success in sentiment analysis systems relies on the model's capability to compose sentiment, the paper aims to identify how non-compositional sentiment meaning is handled by systems. The authors extract a sample of 500 phrases from the SST dataset following certain criteria (phrases consisting of two sub-phrases that contain 3-8 tokens) and then replace one of the sub-phrases with other phrases of the same syntactic type and with the same sentiment. Then the phrases are annotated by annotators recruited via Prolific and annotation is carried out via an online survey system (Qualtrics). Inter-annotator agreement is calculated, returning a good score. The final sentiment labels for phrases are averaged and the non-compositionality ratings are calculated. The final dataset consists of 259 rated phrases.

The results show that most phrases follow the compositionality principle in terms of sentiment, but there are exception, which are usually the result of figures of speech, common sense reasoning, and  discourse relations.

**Reasons To Accept:**

The resources generated by the authors could be useful to other researchers to further explore non-compositionality of sentiment, as well as the evolution of systems when sentiment nuances enter into play. The work is original and goes beyond the usual "turn-of-the-screw, improving-on-benchmarks" paper.

**Reasons To Reject:**

The paper has certain limitations, mostly methodological, which the authors acknowledge and explain in detail, but I don't think these take away any value from the research.

**Reproducibility:**

5: Could easily reproduce the results.

**Reviewer Confidence:**

4: Quite sure. I tried to check the important points carefully. It's unlikely, though conceivable, that I missed something that should affect my ratings.

---

> ### Author Rebuttal · Authors · 2023-08-28
>
> We were glad to hear you value the original approach of our work and the fact that we adequately list the existing limitations of our work; we appreciate the time and effort that went into the reviewing process.

---

### Meta-Review · Area_Chair_YoDo · 2023-09-04

**Recommendation:** 4
**Confidence:** 5

**Metareview:**

The paper explores the role of non-compositionality in sentiment within phrases, aiming to understand how sentiment analysis systems handle phrases whose sentiment doesn't align with expectations. Utilizing a sample of 500 phrases from the SST dataset, the authors conduct a rigorous experiment involving syntactic replacements and human annotations. The annotation process employs an online survey system, Qualtrics, and yields a good inter-annotator agreement score. The final dataset comprises 259 rated phrases, which are then analyzed for their non-compositionality ratings.

The findings reveal that although most phrases adhere to the principle of compositionality in terms of sentiment, exceptions exist. These are typically the result of figures of speech, common sense reasoning, and discourse relations. The paper not only offers valuable insights into the compositionality of sentiment labels but also provides a useful dataset for further research. Written clearly and thoughtfully, the paper presents a novel perspective for evaluating existing sentiment analysis methods.

Overall, despite minor concerns of reviewers, I would still suggest the paper get accepted.

---

### Decision · Program_Chairs · 2023-10-07

**Decision:**

Accept-Findings

**Comment:**

The paper explores the role of non-compositionality in sentiment within phrases, aiming to understand how sentiment analysis systems handle phrases whose sentiment doesn't align with expectations. Utilizing a sample of 500 phrases from the SST dataset, the authors conduct a rigorous experiment involving syntactic replacements and human annotations. The annotation process employs an online survey system, Qualtrics, and yields a good inter-annotator agreement score. The final dataset comprises 259 rated phrases, which are then analyzed for their non-compositionality ratings.

The findings reveal that although most phrases adhere to the principle of compositionality in terms of sentiment, exceptions exist. These are typically the result of figures of speech, common sense reasoning, and discourse relations. The paper not only offers valuable insights into the compositionality of sentiment labels but also provides a useful dataset for further research. Written clearly and thoughtfully, the paper presents a novel perspective for evaluating existing sentiment analysis methods.

Overall, despite minor concerns of reviewers, I would still suggest the paper get accepted.